# Injury profile suffered by targets of antipersonnel improvised explosive devices: prospective cohort study

Shane Smith,[1,2] Melissa Devine,[3] Joseph Taddeo,[4] Vivian Charles McAlister[1,2]

[1]Royal Canadian Medical Service, London, Ontario, Canada
[2]Division of General Surgery, University of Western Ontario, London, Ontario, Canada
[3]Royal Canadian Medical Service, Halifax, Nova Scotia, Canada
[4]Department of Surgery, Maine Veterans' Affairs Medical Center, Augusta, Maryland, USA

**Correspondence to**
Dr. Vivian Charles McAlister; vmcalist@uwo.ca

## ABSTRACT

**Objective** To describe pattern 1 injuries caused by the antipersonnel improvised explosive device (AP-IED) in comparison to those previously described for antipersonnel mines (APM).

**Design** Prospective cohort study of 100 consecutive pedestrian victims of an AP-IED, with traumatic amputation without regard for gender, nationality or military status.

**Setting** Multinational Medical Unit at Kandahar Air Field, Afghanistan.

**Participants** One hundred consecutive patients, all male, 6–44 years old.

**Main outcome measures** The details of injuries were recorded to describe the pattern and characterise the injuries suffered by the target of AP-IEDs. The level of amputation, the level of soft tissue injury, the fracture pattern (including pelvic fractures) as well as perineal, gluteal, genital and other injuries were recorded.

**Results** Victims of AP-IED were more likely, compared with APM victims, to have multiple amputations (70.0% vs 10.4%; p<0.001) or genital injury (26% vs 13%; p=0.007). Multiple amputations occurred in 70 patients: 5 quadruple amputations, 27 triple amputations and 38 double amputations. Pelvic fracture occurred in 21 victims, all but one of whom had multiple amputations. Severe perineal, gluteal or genital injuries were present in 46 patients. Severe soft tissue injury was universal, with injection of contaminated soil along tissue planes well above entry sites. There were 13 facial injuries, 9 skull fractures and 3 traumatic brain injuries. Eleven eye injuries were seen; none of the victims with eye injuries were wearing eye protection. The casualty fatality rate was at least 19%. The presence of more than one amputation was associated with a higher rate of pelvic fracture (28.6% vs 3.3%; p=0.005) and perineal–gluteal injury (32.6% vs 11.1%; p=0.009).

**Conclusion** The injury pattern suffered by the target of the AP-IED is markedly worse than that of conventional APM. Pelvic binders and tourniquets should be applied at the point of injury to patients with multiple amputations or perineal injuries.

## INTRODUCTION

Antipersonnel mines (APM) came into widespread use in the Second World War. Originally they were used to protect antitank mines, but then went on to become a weapon system in and of themselves. They were

## Strengths and limitations of this study

► This is a prospective consecutive series with a predefined definition of the target of the antipersonnel improvised explosive device (AP-IED) attack used as inclusion criteria.
► The definition of the target of the AP-IED was chosen to facilitate comparison to the International Committee of the Red Cross land mine data. This excluded injured bystanders who may have more minor injuries.
► The Role 3 Multinational Medical Unit was the only surgical facility in the region. This maximised catchment of AP-IED victims and minimised selection bias.
► Patients who died at the scene of the explosion may not have been evacuated to the Role 3 Multinational Medical Unit and victims who died after discharge would not have been included in this study. This may have resulted in an underestimation of the AP-IED's lethality. However, studies of antipersonnel mines victims, to which this study is compared, are subject to a similar risk of underestimation.

designed to injure but not kill to remove a target from combat and to increase the logistical burden for the enemy of caring for the casualty. These weapons are typically buried and left for the target to trigger. The weapon is indiscriminate because non-combatants such as children and civilians may detonate them. They are also frequently left buried and active after fighting in that region, or the conflict itself, has ended. Following a public relations campaign that highlighted the indiscriminate injuries cause by APMs, 162 countries signed the 1997 Ottawa treaty promising to cease their production and use.[1] An important basis of that campaign was the clinical description of APM-related injuries by Robin Coupland and Adriaan Korver for the International Committee of the Red Cross (ICRC).[2] Three patterns of injury were found among 754 victims treated at two ICRC hospitals following wounding by blast or fragmentation mines. In pattern 1 injuries, the

victim triggers a blast mine by stepping on it and suffers the full effect of the explosion. In pattern 2 injuries, victims are farther away from the centre of the explosion, whether it is a blast or fragmentation mine and suffer wounds from fragments. Pattern 3 injuries are caused by handling the mine (blast or fragmentation), resulting in severe injuries to the hands and possibly the face and chest. Pattern 1 victims suffer one or more traumatic amputations of the lower limb, whereas pattern 2 victims have fragment wounds scattered over the body and only rarely suffer amputation.

This distinguishing feature of a traumatic amputation between pattern 1 and pattern 2 injuries was confirmed in a later report of 4616 APM victims which presented further evidence of the severity of injury particularly to the limbs.[3] Of 1077 (23%) pattern 1 victims, 606 (56%) had below-knee amputation (BKA), 26 (2%) bilateral BKA, 349 (32%) unilateral above the knee amputations (AKA), 57 (5%) bilateral AKA and 39 (4%) a combination of AKA and BKA.[3] Their description of the typical pattern 1 injury profile caused by an APM, which has been confirmed by other studies, is of a traumatic amputation of the foot or leg with scattered penetrating injuries elsewhere.[2–5]

The improvised explosive device (IED) is increasingly used in modern conflicts including Afghanistan.[6 7] The majority of these weapons have been directed against pedestrian individuals in a similar fashion to APM.[8 9] As with APM, injuries suffered by victims of the antipersonnel IED (AP-IED) depend on whether they were the target of the explosive device or at some distance from the centre of the explosion. It does not depend on the manufacture of the explosive device, industrial as in APM versus improvised as in AP-IED. Indeed elements of conventional explosives may be used to construct an AP-IED. The blast injury depends on the energy transferred by the explosion to the victim. However, it was our impression that pattern 1 injuries caused by AP-IED were significantly worse than those reported for APM.[2] The purpose of this paper is to describe the profile of pattern 1 injury suffered by the target of the AP-IED compared with the target of APM.

## METHODS

The cohort of patients under study was defined as those pedestrian victims of an AP-IED who suffered a traumatic amputation (ICRC pattern 1). To be included in the study, casualties were required to have been reported by the first responder as being the dismounted (pedestrian) victim of an IED explosion and to have suffered a traumatic amputation. Casualties that were non-pedestrian (ie, in a vehicle) or whose situation was unknown were excluded. The study was designed to describe the injuries suffered by 100 consecutive targets of AP-IED attacks who presented to the North Atlantic Treaty Organization Role 3 Multinational Medical Unit (R3-MMU) in Kandahar Air Field (KAF). The hospital received casualties from

the point of injury or via a forward treatment centre.[10] A R3-MMU is the highest level of in-theatre medical care and would be equivalent to a civilian level II trauma centre, as defined by the American College of Surgeons—Committee on Trauma. A specifically designated trauma nurse (MD) and attending surgeons (JT, VM) prospectively collected data for each patient. Data were collected between January 2010 and July 2011 or until data from 100 victims had been obtained. Data collected included: nationality, age, gender, mortality status, specific injuries and imaging results. The level of traumatic amputation, the level of soft tissue injury, the fracture pattern (including pelvic fractures) as well as perineal and gluteal injuries were specifically recorded. Generally, coalition patients were transferred after damage control surgery, whereas local patients also received definitive care at the R3-MMU. Data were collected until the time of first discharge from the hospital. To minimise bias, patients were included sequentially as a consecutive series with a predefined definition of AP-IED target. The R3-MMU was the only surgical facility in the region and would receive all AP-IED victims for care. Patient level data is reported identifying the injuries of each of the 100 victims. The paediatric victims are also presented separately. The data collection was complete with injury data captured for all 100 victims. Categorical data were analysed using the $\chi^2$ test. AP-IED data were compared with previously published APM data.[2] This study was approved by the commander of the R3-MMU KAF and by the Research Ethics Board of the University of Western Ontario (REB # 104124).

## RESULTS

One hundred consecutive casualties with amputations from AP-IEDs were identified and their injuries were described. All of the patients were male. The mean, and median, age of the victims was 25 years. The age range was from ages 6 to 44 years. There were nine patients under the age of 18 (figure 1). Twenty-seven patients were Afghan local nationals and 61 were coalition soldiers (USA, Canada, UK); there were 12 victims whose nationality was not captured (table 1). Eleven patients were dead on arrival and another eight died of their wounds in the hospital, giving a 19% casualty fatality rate. Gender, injury pattern, age or approximate age, was recorded for all victims.

AP-IED victims were more likely than a similar cohort of APM victims to suffer multiple amputations (70.0% vs 10.4%; p<0.001) or genital injury (26.0% vs 13.4%; p=0.007). Seventy victims (70%) sustained multiple amputations: five patients suffered quadruple amputations, 27 had triple amputations and 38 victims had double amputations (figure 2). Two victims had hip disarticulations, 41 had at least one AKA and 56 had at least one BKA (including through-knee and through-ankle amputations). Sixty-five victims had bilateral lower extremity amputations and 37 had at least one upper extremity

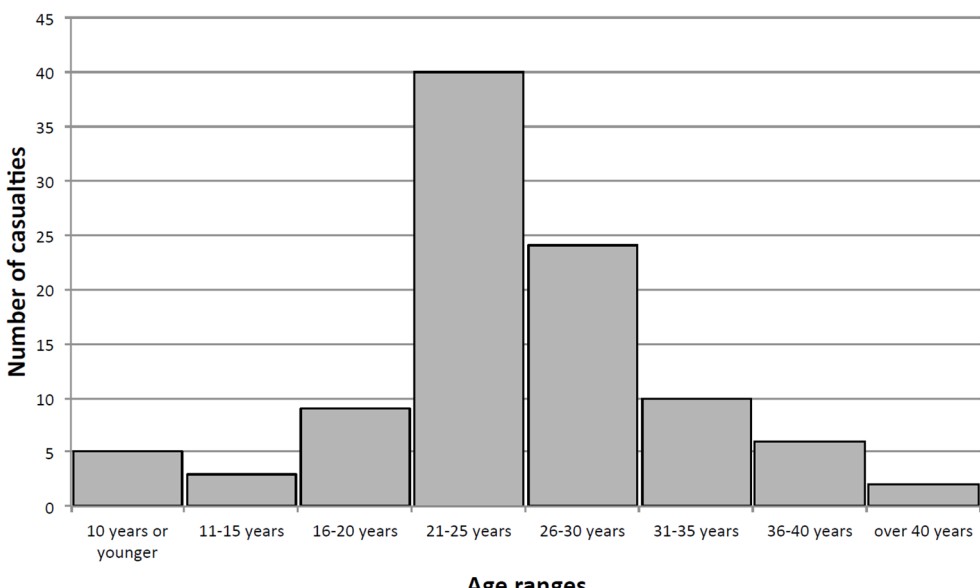

**Figure 1** Age ranges of antipersonnel improvised explosive device targeted victims.

amputation (at the level of the hand or higher) (table 2). Pelvic fractures were present in 21 victims, all but one of whom had multiple amputations. Forty-six patients had perineal and gluteal injuries; these included 8 anorectal injuries, 8 penile injuries and 26 scrotal injuries including 10 orchiectomies. There were 13 facial injuries, 9 skull fractures and 3 traumatic brain injuries. Eye injuries were found in 11 patients, none of whom were wearing eye protection. Of the nine paediatric patients, three had triple amputations and five had double amputations. There was one pelvic fracture and two perineal injuries among the paediatric victims (table 3). Pelvic fracture was more likely if the victim had multiple amputations

(28.6% vs 3.3%; p=0.005) or had a perineal or gluteal injury (32.6% vs 11.1%; p=0.009). Those victims with multiple amputations also had more perineal and gluteal injuries (52.9% vs 30%; p=0.036) and a higher mortality (24.3% vs 6.7%; p=0.039).

Therefore, the typical injury profile suffered by targeted victims of AP-IEDs included: bilateral lower extremity amputations (often above the knee); mangling or amputation of an upper extremity; extensive soft tissue injury with deep contamination by soil, extending into gluteal and perineal regions and pelvic ring disruption and genital mutilation.

| Table 1 | Victim characteristics |
|---|---|
| **Characteristic** | |
| Number of patients | 100 |
| Age, mean (SD) | 25 (6.8) years |
| Median | 25 years |
| Range | 6–44 years |
| Male | 100 |
| Female | 0 |
| Nationality | |
| Afghan | 27 |
| Coalition | 61 |
| American | 49 |
| Canadian | 8 |
| British | 4 |
| Not specified | 12 |
| Mortality | 19 |
| Killed in action | 11 |
| Died of wounds | 8 |

## DISCUSSION

The mechanism of injury is the same for all antipersonnel explosive devices. The severity of injury depends on the energy transferred by the explosion to the victim. There is nothing inherently different between the APM and the AP-IED. Indeed the AP-IED might be considered a subset of APM. However, the comparison in this study is between the conventional blast APM, industrially manufactured for national armies and the AP-IED, that were manufactured locally from available materials and used in the recent conflict in Afghanistan. Coupland and Korver believed they were dealing with the former device.[2–4] In a textbook reflecting on the Russian experience in Afghanistan, Bruysov and colleagues illustrated the mechanism of injury from the conventional blast APM in a series of experiments using rapid sequence photography which confirmed Coupland and Korver's description of a foot amputation with scattered penetrating injuries elsewhere.[11]

The AP-IED, sometimes portrayed as a primitive or crude weapon crafted from locally available resources because of a lack of access to conventional weapons, has

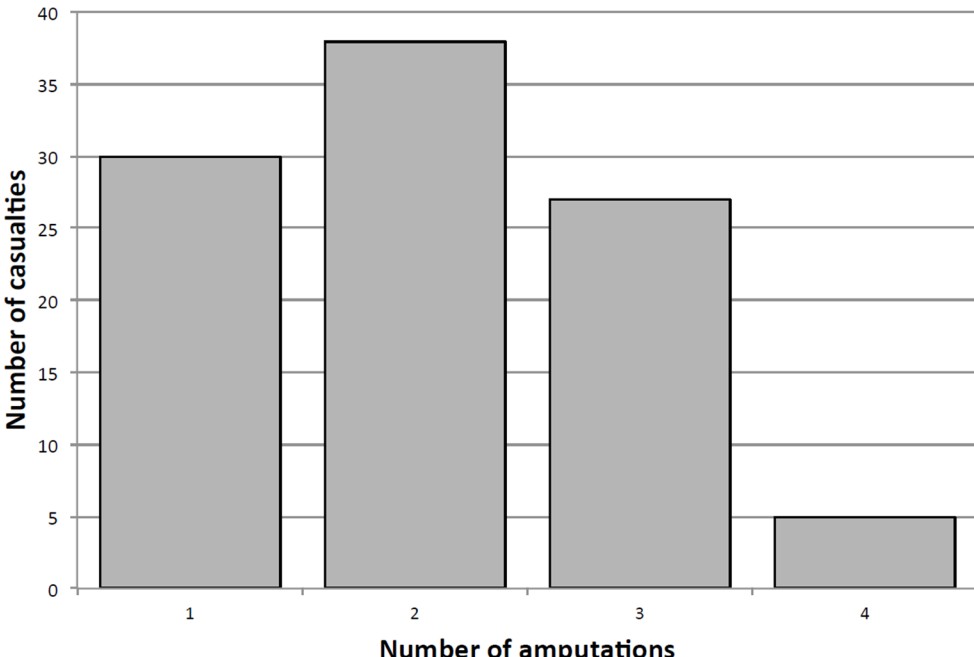

**Figure 2** Numbers of traumatic limb amputations per targeted victim of the antipersonnel improvised explosive device.

evolved with use in different conflicts. From rudimentary devices of nails in wooden boxes used in Cambodia and Columbia, AP-IEDs used in recent conflicts are better directed and more destructive. The injury profile from the modern AP-IED is far worse than originally described for conventional blast APMs. Whereas the blast APM typically results in a unilateral lower limb amputation, the modern AP-IED causes bilateral high amputation of the lower extremities. Triple amputations were not seen with APM but occurred in 27% of victims of AP-IEDs. Severe perineal and gluteal injury with soft tissue contamination with soil was commonly present in the victims of the AP-IED but is not seen among the ICRC experience with APM injured patients.[2] The energy transfer endured by targeted victims of AP-IEDs must be far greater than that caused by APMs. A measure of the force is the fact that it is sufficiently powerful to disrupt the pelvic ring in one-fifth of the patients (figure 3).

It is not the purpose of this paper to undermine the abhorrence of injuries caused by conventional APMs, which are often devastating and ruin lives. Both the APM and AP-IED injure indiscriminately. In this series, almost 1 in 10 victims of AP-IEDs was a child, a tragic feature in common with APMs. Even more disconcerting is the frequency with which children suffer the severest injuries from the powerful explosive force of the AP-IED: 89% suffered multiple amputations and 33% lost three limbs.

The AP-IED victims wore different types of personal protective equipment (PPE) or none at all. Coalition soldiers wore PPE including helmets, body armour and antiballistic eye wear. Each nation issued its own style and brand of PPE. Local Afghan soldiers often used helmets but their use of body armour was more inconsistent and almost never used ballistic eye wear. Civilian casualties

were not wearing any PPE. This heterogeneity of PPE use in our sample may have affected the pattern of injury characterised in this study but the sample size was insufficient to determine accurately the injury preventative role of PPE. We felt there was a lower than expected rate of abdominal, thoracic and eye injuries in victims wearing PPE. We also felt that the design purpose of PPE, to prevent fragment injury, was confirmed in patients with pattern 2 injuries who are not included in this study.

It is possible that we have magnified the severity of the injury pattern by concentrating our focus on patients that met our definition of being targeted by the AP-IED, that is having sustained at least one limb amputation. It is also possible that if a casualty was injured by an unexploded ordinance or classical land mine but reported by first responders to have been injured by an AP-IED, they may have been erroneously included. However, our findings are in keeping with other descriptions of IED injuries in the literature. Jacobs *et al* classified the limb injury pattern of 103 consecutive casualties of IEDs treated at a UK Role 3 facility in Helmand Province, Afghanistan.[12] They included victims who did not have amputation of a limb. They found 76 victims suffered significant bilateral lower limb injuries, with 50 who required bilateral lower limb amputation. Thirty-three victims suffered genital or perineal injury, 9 sustained pelvic ring disruption and 40 sustained significant upper limb injury. They found that all pelvic fractures and 80% of genital injuries were sustained among casualties with bilateral lower limb amputations. A retrospective review of the UK Joint Theatre Trauma Registry examined UK services personnel who were casualties of IEDs in Afghanistan, who sustained lower extremity amputations between January 2007 and December 2010.[13] This registry includes postmortem

**Table 2** Distribution of amputations

| Amputation type | Overall | Pelvic fracture | Perineal, gluteal, genital injury | Killed in action | Died of wounds | Casualty fatality rate |
|---|---|---|---|---|---|---|
| HipDis/TKA/UEA | 1 | 0 | 0 | 0 | 1 | 1/1 |
| HipDis/- | 1 | 0 | 0 | 0 | 1 | 1/1 |
| AKA/AKA/UEA/UEA | 4 | 0 | 3 | 0 | 0 | 0/4 |
| AKA/AKA/UEA | 7 | 3 | 6 | 0 | 0 | 0/7 |
| AKA/AKA | 8 | 2 | 4 | 2 | 1 | 3/8 |
| AKA/BKA/UEA | 2 | 1 | 0 | 1 | 1 | 2/2 |
| AKA/TKA/UEA | 1 | 1 | 0 | 0 | 1 | 1/1 |
| AKA/BKA | 7 | 1 | 5 | 0 | 0 | 0/7 |
| AKA/TKA | 7 | 4 | 6 | 0 | 0 | 0/7 |
| AKA/UEA/UEA | 1 | 0 | 0 | 0 | 0 | 0/1 |
| AKA/UEA | 1 | 0 | 0 | 0 | 0 | 0/1 |
| AKA/TAA | 1 | 0 | 0 | 0 | 0 | 0/1 |
| AKA/- | 2 | 0 | 0 | 0 | 0 | 0/2 |
| BKA/BKA/UEA/UEA | 1 | 0 | 0 | 0 | 1 | 1/1 |
| BKA/BKA/UEA | 13 | 4 | 6 | 4 | 2 | 6/13 |
| BKA/TKA/UEA | 1 | 0 | 1 | 0 | 0 | 0/1 |
| BKA/BKA | 9 | 3 | 5 | 2 | 0 | 2/9 |
| BKA/UEA | 2 | 0 | 0 | 0 | 0 | 0/2 |
| BKA/TKA | 1 | 0 | 0 | 1 | 0 | 1/1 |
| BKA/- | 22 | 1 | 9 | 1 | 0 | 1/22 |
| TKA/TKA/UEA | 1 | 1 | 1 | 0 | 0 | 0/1 |
| TKA/TKA | 1 | 0 | 0 | 0 | 0 | 0/1 |
| TKA/- | 1 | 0 | 0 | 0 | 0 | 0/1 |
| TAA/UEA | 1 | 0 | 0 | 0 | 0 | 0/1 |
| TAA/- | 3 | 0 | 0 | 0 | 0 | 0/3 |
| UEA/- | 1 | 0 | 0 | 0 | 0 | 0/1 |

AKA, above-knee amputation; BKA, below-knee amputation; HipDis, hip disarticulation; TKA, through-knee amputation; TAA, through-ankle amputation; UEA, upper extremity.
Amputation, which includes above elbow, below elbow and hand amputations.

reports of soldiers who were killed and never received hospital care. The registry is restricted to UK services members and contains no local nationals or civilian injuries. They examined 656 IED victims with 138 killed in action (21%) and 31 who died of their wounds (4.7%) resulting in a 25.9% casualty fatality rate. Of the 169 victims who sustained a traumatic lower extremity amputation, 69 were killed in action (40.8%) and 31 died of their wounds (18.3%). They found that the level of amputation was inversely correlated with survival with only two survivors from hindquarter amputation out of 39 patients with this level of injury. Another retrospective review of the UK Military Trauma Registry focused on soldiers from Afghanistan with bilateral leg amputation.[8] They examined 43 casualties with at least two amputations; 80% of these were from AP-IEDs with the other 20% being from antivehicle IEDs. The most common bilateral amputation was a bilateral AKA (58%), and 20% of victims suffered triple amputation. Of the victims, 14% sustained open

pelvic fractures and 44% suffered perineal or genital injury. No victims survived the loss of all four limbs.

The soft tissue injury is a particularly difficult problem encountered by those caring for victims of AP-IEDs. The directed explosion forces soil up along soft tissue planes far above the point of entry. This may worsen the level of eventual amputation and it condemns the victim to multiple operations to remove the contamination. Even still, it may leave the victim at the mercy of unusual antibiotic resistant soil organisms such as *Acinetobacter baumannii*.[14]

Knowing the pattern of injury helps responders tailor care. As Jacobs and colleagues have suggested, first responders should apply tourniquets bilaterally even if the victim is not bleeding, as haemorrhage is likely to start once resuscitation restores intravascular volume.[12] While current protocols require medics to apply a neck collar to all victims of IEDs, spinal injury is rare in AP-IED victims despite the magnitude of the force.[9] On the other hand,

**Table 3** Distribution of paediatric amputations

| Age | Amputation type | Pelvic fracture | Perineal, gluteal, genital injury | Disposition |
|-----|-----------------|-----------------|-----------------------------------|-------------|
| 16 | AKA/BKA | No | No | Survived |
| 14 | AKA/AKA | No | Yes | DOW |
| 14 | AKA/UEA/UEA | No | No | Survived |
| 11 | UEA/TKA/AKA | Yes | No | DOW |
| 10 | BKA/BKA/UEA | No | No | KIA |
| 10 | UEA/TAA | No | No | Survived |
| 10 | BKA/AKA | No | Yes | Survived |
| 10 | BKA/BKA | No | No | KIA |
| 6 | UEA | No | No | Survived |

AKA, above-knee amputation; BKA, below-knee amputation; DOW, died of wounds; KIA, killed in action.; TKA, through-knee amputation, TAA, through-ankle amputation; UEA, upper extremity amputation, which includes above elbow, below elbow and hand amputations.

we recommend that all AP-IED victims with perineal injuries or multiple amputations have pelvic binders placed by the first responders, as both are associated with pelvic fracture. This may reduce the large amount of blood that may be lost silently with pelvic disruption. Multiple surgical teams have to work simultaneously to rescue these patients. Surgeons have to be adept at dealing with pelvic and gluteal bleeding, sometimes by preperitoneal packing and by ligating the internal iliac artery. Rehabilitation requires mobilisation on temporary lower limb prostheses despite injured upper limbs. Finally, victims of this severe injury pattern need life-long support because disabilities may become more pronounced as the capacity to compensate fades with age.

In collecting information on 100 consecutive patients, we hope to have described a generalisable result. Due to the nature of the conflict and the local culture, all of the patients examined were male. While we would expect the severity of the injury pattern to be generalisable to females, we cannot explicitly determine the effect of the modern AP-IED on the female pelvis and perineum.

The use of conventional APM has decreased in the last 20 years since the campaign to prohibit their use. Pattern 1 injuries seen in recent conflicts are far more often from AP-IEDs than conventional APMs. ICRC includes AP-IEDs in the category of APM if the victim triggers the weapon. The injury profile illustrated in the most recent edition of its textbook on war surgery includes those injured by conventional land mines and AP-IEDS.[15] Indeed, there may be overlap in the pattern 1 injury profile of each weapon. For instance, perineal injuries are possible with land mines but are very rare. The difference between the weapons is a matter of the force endured by the victim. Pelvic binders and tourniquets should be applied at the point of injury to patients with multiple amputations or perineal injuries.

Despite the huge force endured by the targeted victim of the AP-IED, the survival rate that we observed was over 80% which is comparable to that reported elsewhere.[15] While the case fatality rate does not account for casualties who were killed at the site of the AP-IED but not brought to the facility or patients that died as a result of injuries after transfer or discharge, the survival rate might be considered greater than expected considering the force endured. The injury pattern suffered by the survivors of the AP-IED is markedly worse than that of conventional APM. It is a weapon which of its nature causes superfluous injury and unnecessary suffering. Just as the reports by Coupland and Korver provided the medical evidence on which the prohibition of conventional APM was based, it is hoped that reports regarding the pattern of injury caused by the modern AP-IED will result in an abhorrence of this weapon and those who use it.

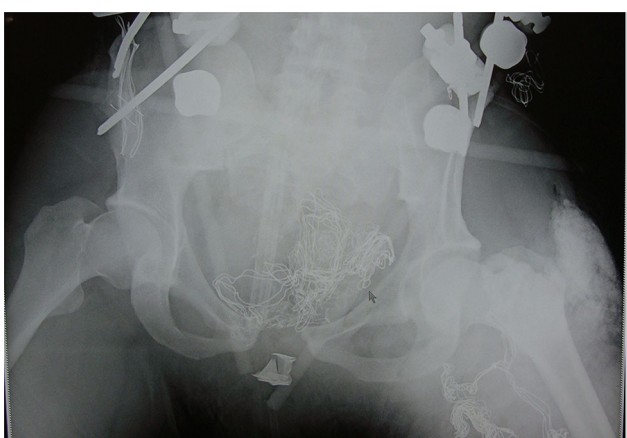

**Figure 3** X-ray taken for placement of a pelvic external fixator showing: disruption of pelvic ring by the force of an antipersonnel improvised explosive device. The arrow indicates combat gauze packing a severe perineal injury; silica laden soil injection by the explosion.

**Acknowledgements** The views expressed in this paper are those of the authors and do not constitute the views or policies of the Canadian Armed Forces or the United States Navy. Transparency declaration: The lead author affirms that the manuscript is an honest, accurate and transparent account of the study being reported; that no important aspects of the study have been omitted and that any discrepancies from the study as planned have been explained.

**Contributors** SS analysed the data and coordinated writing of the manuscript. MD conducted data collection. VM and JT were the treating trauma surgeons of the casualties and designers of the study. They also conducted data collection. The manuscript has been contributed to, reviewed and approved by all authors. All authors had full access to all of the data in the study and can take responsibility for the integrity of the data and the accuracy of the data analysis.

**Funding** This study did not receive any outside funding. The Canadian Armed Forces employed SS, MD and VM at the time of the study. The US Navy employed JD at the time of the study.

**Disclaimer** The lead author affirms that the manuscript is an honest, accurate and transparent account of the study being reported; that no important aspects of the study have been omitted and that any discrepancies from the study as planned have been explained.

**Competing interests** None declared.

**Ethics approval** University of Western Ontario Research Ethics Board.

**Provenance and peer review** Not commissioned; externally peer reviewed.

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
