## [Reviewer comments · BMJ Open]

ARTICLE DETAILS

TITLE (PROVISIONAL)	Injury profile suffered by targets of antipersonnel improvised explosive devices: a prospective cohort study
AUTHORS	Smith, Shane; Devine, Melissa; Taddeo, Joseph; McAlister, Vivian

VERSION 1 - REVIEW

REVIEWER	Jonathan Morrison Royal Centre for Defence Medicine, Birmingham, United Kingdom
REVIEW RETURNED	12-Nov-2016

GENERAL COMMENTS	The authors have presented a description of a series of severely injured patients, who were victims of IEDs in Souther Afghanistan. The study is of interest and the comparison with historic ICRC landmine data is novel, although not well exploited in my opinion. The issue I have with this study, is that it does not present anything that is not already know. I disagree with a new naming convention for IEDs - the term "Dismounted Complex Blast Injury" - is well described and accepted within military circles. I think that the data presented is of value as a record of military surgery, but is more suited to specialist military publication like Military Medicine.
--

REVIEWER	Edward Spurrier Defence Medical Service
REVIEW RETURNED	14-Dec-2016

GENERAL COMMENTS	Overall this is an interesting and useful paper which I think merits publication. However I suggest a few things which I hope will strengthen it significantly. 1. The definition of "target" requires that the victim suffered an amputation. The Red Cross injury pattern description suggests that about 25% of landmine targets suffer an amputation, yet this paper is used as justification for an amputation defining the target of an IED. Were any IED victims seen who did not have an amputation? I think further clarification and justification is required.2. You suggest that IEDs cause similar injuries to road traffic collisions. Other papers suggest that IEDs lead to much more significant injuries than RTC victims. I think this statement needs justification or alteration.3. You mention the effects of PPE. While the points made are likely valid, some statistical support would be useful. If this is not possible
---

	for security reasons, make this clear. 4. There is some description of associated injury patterns. A simple statistical analysis to confirm the significance of these associations would be worthwhile. Minor points p7Line 29-30 "they was replaced" should read "they were.." p15 Line 18-19 "desribces" should be "described"
--	---

REVIEWER	D. C. Covey, MD, MSc, FACS Department of Orthopaedic Surgery University of California, San Diego San Diego, CA, USA
REVIEW RETURNED	26-Dec-2016

GENERAL COMMENTS	This is an excellent paper with data collected under arduous circumstances. It provides important differentiation between anti-personnel landmines and anti-personnel IEDs. These data will be beneficial in planning for logistical support and manning of facilities that will be expected to treat IED injuries
--

REVIEWER	Christos Giannou Associate Professor, Queen Mary & Barts School of Medicine, Blizard Institute, University of London (UK)
	Retired: Head Surgeon, International Committee of the Red Cross, Geneva, Switzerland
REVIEW RETURNED	26-Dec-2016

GENERAL COMMENTS	Giannou comments [Full disclosure: I am the former Head Surgeon of the ICRC and was the Medical Co-ordinator of the ICRC campaign to ban anti-personnel mines (APM). As such, I was involved in the negotiations concerning the UN Convention on Conventional Weapons (1995-96) that banned blinding laser weapons and revised the Protocol on landmines, and then the process that lead to the Ottawa Treaty (1997). I am as well the senior author of the ICRC books War Surgery: Working with Limited Resources in Armed Conflict and Other Situations of Violence, Volumes 1 (2009) & 2 (2013) and wrote most of the contemporary ICRC medical literature concerning APM during the Ottawa negotiations, following on from the work of my colleagues Coupland, Korver, and others. Volume 2 of War Surgery: Working with Limited Resources contains an entire chapter on APM, as well as injuries due to anti-tank mines and to blast and explosive devices in general. It is because of this background that I put into question your definitions, which is the basis of my critique of your paper.] I have several criticisms to make:  1. Your definitions of IEDs
---

2. Your interpretation of the ICRC classification of mine-injury patterns
3. Your inclusion criteria
4. Your treatment of PPE and what this means for the patterns of injury that you describe
5. Your comments on mortality

tend to be classified in defense literature by the type of explosive or delivery system (for example suicide or car bomb). Medically, it is more appropriate to classify IEDs by their target: crowd attack IED (CA-IED); anti-vehicle IED (AV-IED) and the anti-personnel IED (AP-IED). CA-IEDs are designed to injure groups collected

First and foremost, you have confused popular military jargon with technical distinctions that renders your categories problematic. An IED does not constitute a weapon system in and of itself. You would have done well to consult with a military jurist, or one involved in the negotiations of the UN Convention on Conventional Weapons and the Ottawa Treaty on antipersonnel landmines (APM).

The physics of the detonation of all explosive devices (bomb, artillery shell, mortar, grenade, landmine etc.), and their wounding potential, are the same. The “medical effect” – pathogenesis – is identical whether due to an industrially manufactured product or an improvised one. The destructiveness of any explosive weapon depends on the quantity of explosive charge, accompanying metallic fragments, and the environment (open field, closed space etc.) IEDs often contain industrial products (artillery shells, mortars, landmines) in order to increase the force of the explosion and augment the number of metallic fragments, whether from the casing or added secondary fragments (nuts and bolts).

Thus, a tactical definition of a weapon system is not the best medical categorisation. In terms of the physics involved and the medical effects, an IED is simply an ED. (*War Surgery Volume 2 Blast Phenomena*, page 22)

The use of IEDs is not new, furthermore, contrary to contemporary medical literature written by authors from “coalition” forces: CA-IEDs (a car bomb for example) were used extensively in Beirut in the 1970s and 80s (I was there). More important, what is the difference between your CA-IED of 200 kg explosive charge going off in the middle of a crowd and a 200 kg air-dropped bomb exploding in the middle of the same crowd? In terms of the physics and the medical effects: none. The CA-IED is not a “special” weapon.

AV-IED and AP-IED were used by the Afghan mujahedeen against Soviet forces in the 1980s. There is an extensive Soviet literature (I include the most relevant references) that describes in greater detail the wounds that you describe. The effects of a commercial anti-vehicle mine (AVM) and an AV-IED are basically the same. Armour-penetrating weapons (shaped charge for example) have different

effects.

More to the point: what is the difference between an AV-IED and an AP-IED? According to you, whether the victim was “dismounted” or in a vehicle. This is a description of the tactical situation and not the weapon. You are dealing with the same weapon and, therefore, the categorisation of AP-IED or AV-IED is specious. What is the difference between an AVM and an APM? The amount of pressure required to detonate the mine and the quantity of explosive charge. APMs can contain as little as 30 gm of explosive; AVM contain 10 kg and more of explosive in order to put a tank out of action. The weight of an individual is insufficient to cause an AVM to detonate. Therefore, the AVM and APM are distinct weapons. Your AV-IED and AP-IED are apparently the same weapon; the only difference being whether the target is on foot or not. Thus, the victim is exposed to a very large explosion. No wonder the injuries are severe.

On the other hand, the mechanism of detonation is essential in defining a landmine. A landmine – anti-personnel or anti-vehicle – is defined by the tactical situation: it is set off by the target-victim(s). If a combatant is involved – pushing a button, using a mobile phone to detonate – it is not considered a “landmine” because, supposedly, the combatant sees the potential victim and is able to discriminate between military and civilians. The distinction is best exemplified by the Claymore directional weapon, which can be set off by a tripwire (landmine) or by pushing a button (remote control). It is the tactical situation of the mechanism of detonation that defines whether the weapon is a landmine or not. It does not matter how the landmine is constituted: commercial industrial production or improvised. The Ottawa Treaty describes both and covers both. It is the victim that provokes the detonation of an APM, and this is the legal definition of an APM.

One problem for ICRC data collection, as well as that of everyone else, is that it is impossible to differentiate between an APM and unexploded ordnance (UXO), especially unexploded cluster munitions. One cannot tell from the injury whether the cause was an APM or an UXO. It is the local environment that sometimes help in differentiating the two, and the only way to do this is through interviews of all the casualties, and any non-injured bystanders. The ambiguity has always been made clear in the literature dating from the 1990s onward (the campaign to ban landmines). Your references are out of date and my colleagues (Robin Coupland, Korver and others) recognised this problem later on as we gathered more information in the ICRC. These interviews and questionnaires are the mainstay of landmine victim databases. They also help in determining, more or less, the mortality rates, especially for all those who die before entering a medical system and are not calculated in hospital statistics. You state this problem, but in your data collection you do not mention any interviews of casualties or uninjured witnesses.

You start the Discussion section with the statement: “the AP-IED has been likened to an improvised antipersonnel landmine”. Not only has it been likened, technically it is, if the victim sets it off and not a combatant by command at a distance. In your Methods section you state: “casualties were required to have been reported as being dismounted”, but you do not deal with the mechanism of triggering the explosion: pressure plate, trip wire, or remote control? This is important, especially if you want to use legal treaties to ban AP-IEDs.

Some improvised APM (AP-IED) are very rudimentary (the Khmer Rouge and the FARC in Colombia made many out of wooden boxes filled with explosive and nails and even faeces); others have incorporated industrial products (artillery or mortar shells, grenades etc.) and various metal objects in order to increase the power of the explosion and the production of secondary fragments on explosion, which has been the case in Afghanistan and Iraq.

To resume: your CA-IED is simply a bomb going off in a crowd of persons (nothing new); the AV-IED is an anti-vehicle landmine, again with the proviso that it is the target-victim that sets it off and not the voluntary will of a combatant (AVM are quite legal according to current international treaties). And the AP-IED is simply an APM, if the victim provokes the detonation and, as such, it is already illegal according to the Ottawa Treaty. Apparently, your AP-IED is a very large APM.

Secondly, your description of the ICRC patterns of APM injury is incorrect. The ICRC classification is based on the fact that there are two basic types of APM: blast and fragmentation. The blast or contact APM is detonated by the victim stepping on a pressure plate (Pattern 1); the fragmentation mine is set off by means of a tripwire (Pattern 2). Pattern 3 is due to manipulation of a mine, and both types of mine will cause amputation of the fingers, hand, or arm, as well as injuries to the face and chest.

The contact APM invariably causes a traumatic amputation of the limb that enters into contact because of the blast effect; the level of the amputation – and any further injury to the other leg, perineum, abdomen, contralateral upper limb (you mention elbow protection, but this is seen with blast mines because the contralateral arm is swung out over the contact foot when walking) – depends on the quantity of explosive charge in the mine. The results vary from amputation of only part of the foot to bilateral lower limb amputations, to death.

Pattern 2 injuries are those suffered from any fragmenting device (grenade, mortar shell etc.). Indeed, some fragmentation mines are simply hand grenades with a tripwire attached. Fragmentation mines can also cause amputation of a limb, and often do, although not as frequently as with blast mines. It depends on the exact type of mine and, especially, the distance of the victim from the mine at the time of deflagration. If close enough, then the fragments hit the body

within the radius of the primary blast effect with much greater wounding potential. The same ballistic phenomenon is found with any type of fragmentation device: artillery shell, mortar, bomb, grenade etc. (Note the large number of amputations amongst the victims of the Boston marathon bombing: a CA-IED in your terms, but a similar weapon system to a fragmentation APM = fragmentation explosive device.) By the way, the ICRC landmine database (I was personally involved in setting up the definitive version.) includes any injured persons who may have been accompanying the primary victim – the one who set off the mine. Your exclusion of “injured bystanders” is therefore not justified for comparison purposes.

Pattern 3 injuries, manipulation of a mine: obviously, any mine, indeed any explosive device, will cause an amputation of the fingers, hand, or arm, and injuries to the face and chest because of the proximity of the explosion. These injuries are only seen, again obviously, in the presence of very small mines (such as the Soviet PFM “butterfly” mine, used extensively in Afghanistan); larger mines kill the manipulator.

“predefined definition of the target of the AP-IED attack”

The final problem of definitions that I have with your paper is that you define an AP-IED victim *a priori* as a casualty who has suffered at least one lower limb amputation (but you do not know if he encountered an AP-IED, a classical APM, or an UXO). Rather than describing the injuries of all victims of an AP-IED you have predefined the pattern description. The logic is a circular tautology: it is useless as a study methodology. In your discussion, you cite Jacobs et al. whose series included victims of IEDs who did not suffer an amputation, thus undercutting (should I say “undermining”?) your definition, and rightly so.

A further methodological problem is the use of advanced body armour (PPE). This you mention only in your discussion, but it should have been analysed earlier in the Methods and Results section. The injuries sustained by civilians or Afghan soldiers – not wearing such protection – should then have been analysed quite separately from those soldiers who were so protected and not lumped together. You mention certain differences, but the two categories should have been analysed in separate Tables, and then compared.

The pattern you describe does not indicate the true distribution of injuries, rather the distribution of those suffered by (mostly) soldiers wearing PPE. How many did not die because they were wearing body armour? In effect, your study shows the results of a high-explosive charge “bomb” with a high percentage of amputations amongst survivors because PPE prevented lethal torso injuries. The results show the military prejudice of industrial country armies that can afford PPE, but therefore cannot be considered “universal”.

Civilians and poor-country military do not have such PPE. By the way, aerial bombardment can give a similar clinical picture (multiple amputations, severe soft tissue damage, pelvic and perineal injuries etc.: I have pictures from Gaza. Your armed forces, and those of other coalition countries, have not suffered such casualties from aerial bombardment since the Second World War and, therefore, you are unaware of these types of injuries.)

What you can discuss is the radical change in the anatomic distribution of wounds that has occurred due to the use of advanced PPE. This is true of both Iraq and Afghanistan and a number of military colleagues have brought attention to this phenomenon, especially due to the large number of survivors suffering from blast traumatic brain injury and the long-term sequelae.

Mortality: in the Methods section, you mention in-field deaths, but not post-operative mortality after transfer. This is stated only in the Discussion, but should have been mentioned much earlier in the Methods as well. You could then discuss the question with reference to the many articles on APM epidemiology that deal with this problem of unreported deaths: one of the main arguments of the campaign to ban APM and which showed up in the epidemiological surveys of survivors, witnesses and families.

Your description of the pathology is quite correct, if incomplete. You will find a similar but more extensive description in *War Surgery Volume 2 chapter on APM injuries*; the blast effect of an APM (or AP-IED) can also cause intimal damage of major vessels resulting in a propagating thrombus that calls for an even higher amputation than what is thought to be the case on first sight.

Your conclusion concerning superfluous injury and unnecessary suffering is correct, but that is because an AP-IED triggered by the victim is an anti-personnel landmine and is already banned. If set off by a combatant, it is simply a large bomb and, if you want to ban this weapon because of its severe wounding effects, then you would also have to ban 200 kg (and heavier) aerial bombs; not likely to happen anytime soon.

I am open to answering any questions that you may have. I suggest a reworking of your study, which includes much important information that should be reported. I hope that you have retained the medical records of the other non-amputee injured patients so that your study resembles that of Jacobi et al.

My e-mail address is: x.giannou@gmail.com

References include:

Nechaev EA, Gritsanov AI, Fomin NF, Minnullin IP, eds. *Mine Blast Trauma: Experience from the War in Afghanistan*. St Petersburg: Russian Ministry of Public Health and Medical Industry, Vreden Research Institute of Traumatology; 1995. [English translation: Khlunovskaya GP, Nechaev EA. English publication: Stockholm: Council Communications; 1995. (The translation is a bit stodgy and

	difficult at times, but still quite readable.) Bryusov PG, Shapovalov VM, Artemyev AA, Dulayev AK, Gololobov VG. Combat Injuries of Extremities. Moscow: Military Medical Academy, GEOTAR; 1996. [Translation by ICRC Delegation Moscow] (I include a Word document.) Baskin TW, Holcomb JB. Bombs, mines, blast, fragmentation, and thermobaric mechanisms of injury. In: Mahoney PF, Ryan JM, Brooks AJ, Schwab CW eds. Ballistic Trauma: A Practical Guide, 2nd ed. London: Springer-Verlag; 2005: 45 – 66. Champion HR, Holcomb JB, Young LA. Injuries from explosions: physics, biophysics, pathology, and required research focus. J Trauma 2009; 66: 1468 – 1477. You will also find many references in War Surgery: Working with Limited Resources, Volume 2 The reviewer also provided a files in addition to these comments. Please contact the publisher for full details.
--	--

VERSION 1 – AUTHOR RESPONSE

Reviewer: 1

Reviewer Name: Jonathan Morrison

Institution and Country: Royal Centre for Defence Medicine, Birmingham, United Kingdom

Please state any competing interests or state 'None declared': Non Declared

Please leave your comments for the authors below

The authors have presented a description of a series of severely injured patients, who were victims of IEDs in Souther Afghanistan. The study is of interest and the comparison with historic ICRC landmine data is novel, although not well exploited in my opinion. The issue I have with this study, is that it does not present anything that is not already know. I disagree with a new naming convention for IEDs - the term "Dismounted Complex Blast Injury" - is well described and accepted within military circles. I think that the data presented is of value as a record of military surgery, but is more suited to specialist military publication like Military Medicine.

REPLY: Thank you for your comments with respect to our manuscript. Although much useful publication and exchange has occurred in the military medical literature the problem of IED injuries is not confined to the military community. Civilians are at risk of injury from AP-IEDs. Civilian physicians, local and NGO, have to treat these patients both at the time of injury as well as during their ongoing care and rehabilitation. The injuries recorded in this series are at an order of magnitude worse than those reported 25 years ago in the BMJ. There is a danger the use of AP-IED will proliferate, not only in conflict zones but also at home. Therefore we feel the discussion of these injuries has important medical and societal value.

Reviewer: 2

Reviewer Name: Edward Spurrier

Institution and Country: Defence Medical Service

Please state any competing interests or state 'None declared': None declared

Please leave your comments for the authors below

Overall this is an interesting and useful paper which I think merits publication. However I suggest a few things which I hope will strengthen it significantly.

1. The definition of "target" requires that the victim suffered an amputation. The Red Cross injury pattern description suggests that about 25% of landmine targets suffer an amputation, yet this paper is used as justification for an amputation defining the target of an IED. Were any IED victims seen who did not have an amputation? I think further clarification and justification is required.

REPLY: We have revised the description of the cohort studied to align it with the ICRC pattern 1 injury. Coupland and Korver describe pattern 2 injuries as those suffered by a victim a little distance from the centre of the explosion. The ICRC War Surgery textbook states that these injuries are caused by trip-wire ignition. Therefore they too could be defined as a target. We chose the former definition because it also covers those injured when a person close to them steps on a pressure-plate to trigger an IED.

2. You suggest that IEDs cause similar injuries to road traffic collisions. Other papers suggest that IEDs lead to much more significant injuries than RTC victims. I think this statement needs justification or alteration.

REPLY: We have deleted that section because it is distracting. Thank you for noticing.

3. You mention the effects of PPE. While the points made are likely valid, some statistical support would be useful. If this is not possible for security reasons, make this clear.

REPLY: We have revised the discussion regarding PPE adding as much information as we have. Our study does not have sufficient power to determine with certainty an effect of PPE. We believe the anti-fragmentation effect of PPE has greater effect on Pattern 2 injuries than might be expected for Pattern 1 where the force usually ascends from below.

4. There is some description of associated injury patterns. A simple statistical analysis to confirm the significance of these associations would be worthwhile.

REPLY: Thank you for the suggestion. We have added statistical analyses to illustrate the claim that injuries from this weapon are worse than landmines. We undertook internal analyses to compare the impact IEDs that cause a single amputation with those that cause multiple amputations. We show an association of multiple amputations with pelvic disruption. This finding is helpful because it reinforces the notion that first responders should apply a pelvic binder at the point of injury to reduce blood loss.

Minor points

p7Line 29-30 "they was replaced" should read "they were.."

p15 Line 18-19 "desribces" should be "described"

REPLY: Corrected or deleted. Thank you for noticing.

Reviewer: 3

Reviewer Name: D. C. Covey, MD, MSc, FACS

Institution and Country: Department of Orthopaedic Surgery, University of California, San Diego, San Diego, CA, USA Please state any competing interests or state 'None declared': None declared

Please leave your comments for the authors below

This is an excellent paper with data collected under arduous circumstances. It provides important differentiation between anti-personnel landmines and anti-personnel IEDs. These data will be beneficial in planning for logistical support and manning of facilities that will be expected to treat IED injuries

REPLY: Thank you very much for your review and comments.

Reviewer: 4

Reviewer Name: Christos Giannou

Institution and Country: Associate Professor, Queen Mary & Barts School of Medicine, Blizard Institute, University of London (UK); Retired: Head Surgeon, International Committee of the Red Cross, Geneva, Switzerland

Please state any competing interests or state 'None declared': None declared

Please leave your comments for the authors below

Check list 9 (results) and 10 (presentation of results): really depend on number 1 (research question). The paper requires major revision. Please see my comments in the attached document.

** The application was unable to attach manuscript files to this email, because one or more of the files exceeded the allowable attachment size (10MB). **

REPLY: We have great respect for the work Dr Giannou has done for victims of war, injured by a wide array of weapons. We agree with him, and we have stated in the introduction and the discussion, that the AP-IED is only a type of anti-personnel mine. We have completely revised the introduction to remove any element that has not been previously described. We clarified that we sought to describe Pattern 1 injuries caused by IEDs. With respect to 'interviews', subjects were only included in the series if the patient, a witness or the first responder reported that the injuries were caused by an IED and the victim was pedestrian. This information was part of our normal practice and the series was collected prospectively. In the discussion, which is also revised, we used the advice of the reviewer to focus the review. The Russian experience and the ICRC War Surgery textbook provided by Dr Giannou were very helpful. The former included rapid sequence photo-analyses to study the mechanism of landmine injury. The latter describes the devastating injuries and outlines the care for patient injured by anti-personnel explosive devices, industrially made or improvised. It also provided a table of case-fatality rates previously reported as a context for this report.

We believe that we have accommodated all of the reviewer's suggestions, with the possible exception that we stuck with the Coupland and Korver definition of Pattern 2 injuries because it is more inclusive than that in ICRC War Surgery. This is a less relevant issue as we all agree on the definition of Pattern 1 injuries which were the focus of the study.

VERSION 2 – REVIEW

REVIEWER	Christos Giannou Queen Mary, University of London: Masters in Trauma Sciences Personal address: Monemvasia, Greece
REVIEW RETURNED	04-May-2017

GENERAL COMMENTS	The second draft of this paper is much improved. Many suggestions have been taken into account. There are still a few problems in definitions. The authors stray into the territory of international humanitarian law, the laws of war, without the requisite expertise, however. They use the term "superfluous injury and unnecessary suffering", the English
--

	version of the original French "maux superflus" (first mentioned in the St Petersburg Declaration of 1868). Here, and elsewhere in the paper, (especially concerning certain definitions) I suggest that they consult with a military jurist. I include in attachment a revision of their paper with some suggestions. Should the authors agree to these changes, then I would have no problem in accepting their paper for publication. The reviewer also provided a marked copy with additional comments. Please contact the publisher for full details.
--	---

VERSION 2 – AUTHOR RESPONSE

We have tried to accommodate the reviewer with the changes made to this revision.

We are military surgeons. It is our opinion that the pain and suffering documented here qualifies for the moderate comments made in the discussion. We have only used standard definitions, each of which is referenced. We have not made any overt comments regarding the legality of this weapon. We believe that falls to others to do. All we wish to do is objectively document the injury pattern. Therefore we have declined to recruit a military jurist for this paper because we think it is beyond its scope. However we agree with the reviewer that the topic is worthy of a jurist's assessment in later papers.

VERSION 3 – REVIEW

REVIEWER	Christos Giannou Honorary Lecturer, Queen Mary University of London, Blizard Institute Former Head Surgeon, International Committee of the Red Cross
REVIEW RETURNED	06-Jun-2017

GENERAL COMMENTS	This is an excellent article describing the results of an horrific weapon. The recommendation for the pre-hospital application of a tourniquet and pelvic binder is more than justified. Furthermore, the authors have wisely decided not to enter the domain of international humanitarian law concerning the status of IEDs. My present criticisms deal only with certain details and nuances, which do not undermine the basic thrust of the article. 1) The Ottawa Treaty entered into force in 1999. It is customary to refer to treaties according to the date that they were negotiated and signed: therefore, the 1997 Ottawa Treaty. The treaty enters into force only when a required number of countries have ratified it, which is different from signing. Some countries sign, but never ratify. 2) Pattern 2 APM injuries can cause amputations, but not very often. This is brought out in later ICRC publications that have studied larger numbers of patients. 3) Pattern 1 APM injuries can involve the perineum and even penetrate into the abdomen; simply not very often. The pattern 1 AP-IED injuries are much more severe, true. Again, further ICRC experience with APM injuries has demonstrated this. 4) The references are up to date and appropriate, but their use is not. I find it strange that several references are given to work by surgeons of the ICRC, yet the authors of the article prefer to refer not to the most recent but to the much earlier ones when dealing with definitions. Perhaps the ICRC refined its definitions because of its own data collection and internal studies and analysis. As with
---

	most medical classification systems, the most recent is the one that is authentic and acceptable.
--	---

VERSION 3 – AUTHOR RESPONSE

We accept all the recommendations. The following revisions were made:

- 1) Title changed as advised. Injury profile suffered by targets of antipersonnel improvised explosive devices: a prospective cohort study
- 2) Reviewer pt 1: Ottawa treaty changed to 1997 as requested. "the 1997 Ottawa treaty promising to cease their production and use"
- 3) Reviewer pt 2: Introduction changed "whereas pattern 2 victims have fragment wounds scattered over the body and only rarely suffer amputation."
- 4) Reviewer pt 3: discussion revised by adding the following: "The injury profile illustrated in the most recent edition of its textbook on war surgery includes those injured by conventional landmines and AP-IEDS.¹⁵ Indeed, there may be overlap in the pattern 1 injury profile of each weapon. For instance, perineal injuries are possible with landmines but are very rare. The difference between the weapons is a matter of the force endured by the victim." The last two paragraphs were revised to eliminate repetition.
- 5) References 15, see #4 above, is to the most recent textbook by ICRC on the topic

Thank you for your help in making this a better paper.